# Enhancing the scalability of distance-based link prediction algorithms in recommender systems through similarity selection

**Zhan Su**\*[☯], **Zhong Huang, Jun Ai**[iD][☯]\***, Xuanxiong Zhang, Lihui Shang, Fengyu Zhao**

School of Optical-Electrical and Computer Engineering, University of Shanghai for Science and Technology, Shanghai, P.R. China

☯ These authors contributed equally to this work.
\* suzhan@foxmail.com (ZS); aijun@outlook.com (JA)

**Data Availability Statement:** The experiment data set used in the paper can be found at the following url. https://grouplens.org/datasets/movielensThe data that support the findings of this study are

## Abstract

Slope One algorithm and its descendants measure user-score distance and use the statistical score distance between users to predict unknown ratings, as opposed to the typical collaborative filtering algorithm that uses similarity for neighbor selection and prediction. Compared to collaborative filtering systems that select only similar neighbors, algorithms based on user-score distance typically include all possible related users in the process, which needs more computation time and requires more memory. To improve the scalability and accuracy of distance-based recommendation algorithm, we provide a user-item link prediction approach that combines user distance measurement with similarity-based user selection. The algorithm predicts unknown ratings based on the filtered users by calculating user similarity and removing related users with similarity below a threshold, which reduces 26 to 29 percent of neighbors and improves prediction error, ranking, and prediction accuracy overall.

## Introduction

Due to the rapid spread of the Internet, which has resulted in a tremendous number of information appearing every second, people have entered an era of information overload. Thus, overwhelming information makes it difficult for consumers to find information that they are interested in. Consumers and merchants both hope to find useful information on the Internet, whether it's about a suitable product or a group of potential interested clients. Excessive data, on the other hand, is a severe impediment to reaching this aim in a harmonic manner, and the current epidemic forces people to turn to the Internet for consumption, learning, and communication, exacerbating the problem.

Fortunately, scientists and engineers have devised systems that automate the calculation of user preferences for various goods and information based on people's existing consumption and evaluation data, and then recommend to them automatically goods or information that they are likely to like but have yet to discover or consume. The whole process is to predict the weights of links and links between users and items in a recommender system, hence, it is also often referred to as link prediction [1, 2].

available. https://github.com/PlayerAI/plosone2022.

**Funding:** Zhan Su received the funding by the Young Scientists Fund of the National Natural Science Foundation of China (Grant No. 61803264). The funder' website is at http://www.nsfc.gov.cn/. The funders had no role in study design, data collection and analysis, decision to publish, or preparation of the manuscript.

**Competing interests:** NO authors have competing interests.

This is the primary function of the recommender system, and it not only saves the user time in searching for an interesting target but it also identifies a possible user group for the merchant. Therefore, link prediction and recommendation algorithms are effective ways to streamline excessive information, saving users time searching for products and information while also saving businesses money on advertising [3, 4].

Thus, the link prediction and recommendation algorithms have been intensively investigated in related fields due to its commercial value and research significance [5]. Amazon, Facebook, JD, Taobao, and other companies have their own personalized recommender systems and algorithms [6, 7].

According to the general understanding of the area, link prediction in recommender systems and recommendation algorithms can be separated into three primary types. Content-based recommendation algorithms, collaborative filtering recommendation algorithms, and hybrid recommendation algorithms together constitute the classification.

First, in content-based recommender systems, the descriptive attributes of items are used to make recommendations. The term "content" refers to these descriptions [8, 9]. Second, memory-based collaborative filtering algorithms and model-based collaborative filtering algorithms are two sub-types of collaborative filtering algorithms.

Memory-based collaborative filtering algorithms [10], whether item-based [11–13] or user-based [14], are based on the assumption that similar individuals demonstrate similar patterns of rating behavior, and similar objects receive similar ratings [15]. On the other hand, clustering models [16], maximum entropy models [17], matrix decomposition models, and similarity-network models [18–21] are examples of model-based collaborative filtering techniques [22, 23].

Finally, there are weighted type, switching type, cross type, feature combination type, waterfall type, feature enhancement type, and meta-level type hybrid recommendation strategies [24].

Despite the fact that the recommender system has been under development for a long time, some issues remain, such as cold start [25], data sparsity, insufficient scalability [26], accuracy improvement, computational cost, prediction vulnerability [27] and lack of diversity [28].

The most pressing issue remains to improve the accuracy of recommendations. To increase the accuracy of the recommendation system, many academics have proposed improved similarity computation algorithms [29]. For measuring similarity, Hawashin et al. used user interests [30]. PK Singh and others consider consumers' likes and dislikes of similar features of a single item separately when determining similarity [31]. Ai et al. [32] used similarity to model user-user network, and considered centrality measures [33] as a factor to enhanced prediction accuracy. By introducing a specified distance function, MA Yi et al. increased Pearson similarity [34]. N Joorabloo et al. proposed a new approach of user/item neighborhood reordering that takes into account the future trend in user/item similarity [35]. All of these techniques increase the accuracy of recommendations based on user similarity prediction.

Other approaches, such as the Slope One algorithm [36] and its variants, treat rating differences as a distance between users, and use that distance to anticipate unknown ratings for the recommendation. LY Dong et al. [37] improved this approach by incorporating non-negative matrix factorization into Slope One, which effectively handled the sparsity problem. W Li et al. [38] created the Slope One algorithm, which improves prediction accuracy and reduces sparsity by using weighted essential items. L Jiang and colleagues [39] presented the Slope One method combined trusted data with user similarities. The algorithms improved the prediction accuracy by using all the possible neighbors.

The Slope One method is not only simple to implement, but also highly effective. The Slope One algorithm's prediction accuracy, on the other hand, could be improved further. This

algorithm's computational complexity is also higher than that of other algorithms. Although there are methods that aim to merge distance and similarity in previous studies, the algorithm's time complexity is increased [39]. Because the Slope One algorithm requires all eligible users in its prediction, the large number of neighbors severely affects its scalability despite its computational simplicity. In practical applications it can save storage space and speed up prediction if only a small portion of neighbors can be cached for prediction.

To solve the problem of Slope One using too many neighbors in the prediction, we present a link prediction method in recommender system based on user difference and similarity selection, which is primarily based on two assumptions. 1) The user-rating distance in recommender system can be measured and used for unknown rating prediction. 2) If the similarity between two users is less than a certain threshold, their rating distance is not helpful for accurately measuring the differences between the users. Therefore, it can be discarded in the calculation of user-rating distance.

The following are the primary contributions of the work.

1. We designed a rating prediction method based on user similarity selection, which reduced more than 40% of neighbors in the prediction.

2. Second, the method successfully decreases the time complexity, improve the overall performance in prediction error, ranking of recommendation list and prediction accuracy of user preference.

3. Third, our study suggests that, neighbors who are not similar enough to the prediction target cause more harm than good to the prediction and should be discarded.

The rest of the paper is laid out as follows: Section II reviews related research; Section III details the suggested approach; Section IV compares the algorithm established in this study to several state-of-the-art algorithms in the field; and Section V summarizes the conclusion and proposes potential future studies.

## Related works

### Measurement of user similarity

Among many others, the collaborative filtering recommendation algorithm is widely explored and utilized. The primary approach is to offer recommendations based on the tastes of comparable user groups to the target user [40]. The algorithm calculates the similarity between the target and all other users, selects a group of highly similar neighbors, and uses the neighbors' ratings to estimate the target user's rating on an item [5].

The calculation of similarity is the most important phase in collaborative filtering algorithms. As a result, many methods in the field of recommender systems have been proposed to solve this problem, such as the Pearson correlation coefficient [41], cosine similarity [5], Ou Kilid distance [10, 42], similarity with confidence measures [43], mean square distance [44], user behavior probability [45], Jaccard [46, 47], Spearman correlation [48], vector similarity [2], Bhattacharyya coefficient [49], and user opinion spreading [50, 51].

### Rating distance between users

The Slope One [36] algorithm is a model-based recommendation technique for calculating average user distance. Its fundamental idea is to employ a linear regression analysis method based on the user's previous movie ratings. Between the user $U$ and $V$, the rating on the same item has a linear relationship, $U = V + b$. The premise of prediction is to predict unknown ratings based on the difference between users. Assume that user $u$ has given the movie $i$ a rating

but user $v$ has not rated the item. $U = V + b$ calculates the deviation value between two users, with $b$ being the deviation value between them. It is important to identify the two users that have jointly rated a group of items in order to calculate the deviation value between them. The average deviation of the ratings between the two users is estimated to be $b$ [5, 36] and used for prediction when one of $U$ and $V$ is unknown.

## The proposed method

### Symbol description

Assume that a total of $m$ users have rated $n$ movies, it can be regarded as a $m \times n$-level rating matrix $R_{m \times n}$, which is represented by the user set $U = \{u_1, u_2, \ldots, u_m\}$, the collection of movies is represented as $I = \{i_1, i_2, \ldots, i_n\}$, $r_{ui}$ represents the rating of the movie $i$ by the user $u$.

The data set is divided into a training set *TrainingSet* and a test set *TestSet*. We use $d_{uv} = dev(u, v)$ to represent the average of the rating distance between user $u$ and $v$, $s_{uv} = similarity(u, v)$ stands for the similarity between users $u$ and $v$, and $T_h$ is a similarity threshold, by which select neighbors for users with similarity larger than or equal to $T_h$.

### Algorithm description

The problem with collaborative filtering is that it does not consider the impact of distance between users, and the traditional distance method does not consider the influence of similarity on user prediction. In addition, the collaborative filtering algorithm needs to calculate and select neighbors for the target user, and the time complexity of the algorithm is relatively higher than distance approach. The issue can be demonstrated by Fig 1. In terms of rating distance, these two user pairings are near ($-0.5$ against $-0.25$), yet their similarities are absolutely opposite ($-1.0$ against $1.0$).

This research provides a link prediction algorithm based on user distance and similarity selection (DSS) to address the aforementioned issues. As shown in the Fig 3, there are three

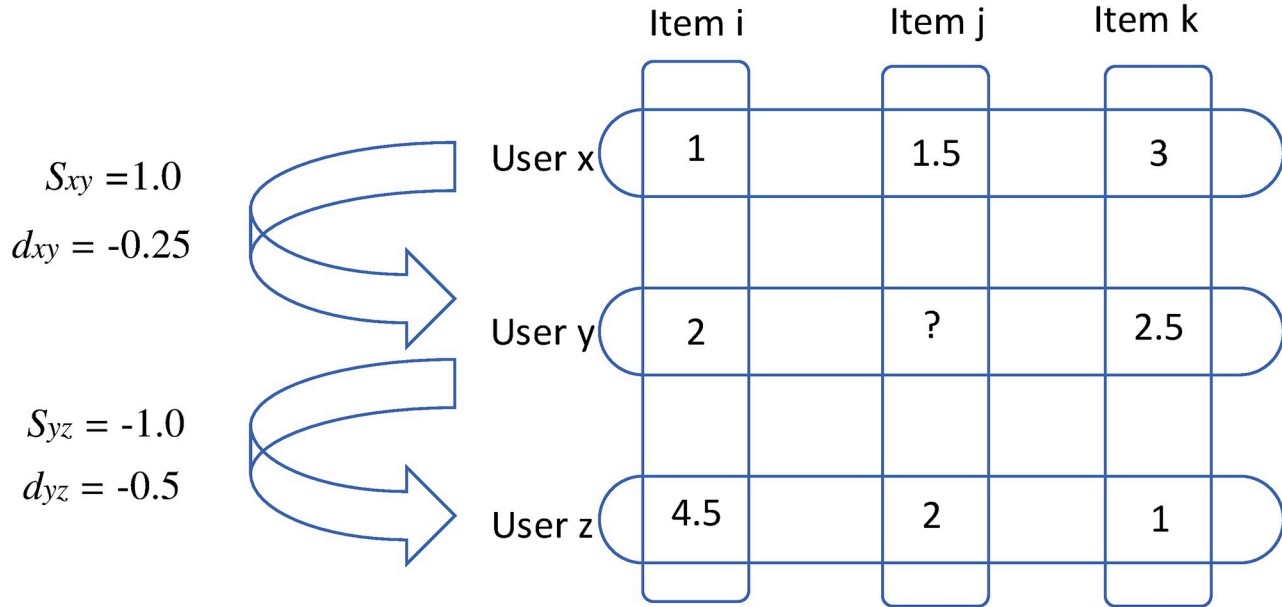

**Fig 1. Ratings of three movies by three users are shown in a schematic diagram of traditional algorithms.** $s$ is given by similarity measurement and $d$ is calculated by distance method.

users $x$, $y$ and $z$, respectively. For the rating $i$, $j$, and $k$, it is necessary to calculate the predicted rating of user $y$ on movie $j$. According to the distance equation, the distance between user $x$ and $y$ calculated as $d_{xy} = -0.25$, and the distance between user $y$ and $z$ as $d_{yz} = -0.5$, respectively. By contrast, the Pearson similarity equation evaluates the similarities between user $x$, $y$ and user $y$, $z$ as $s_{xy} = 1.0$ and $s_{yz} = -1.0$, respectively.

According to our approach, only users with high similarity are selected for the prediction, that means user $z$ is excluded. We use the rating of user $x$ to predict the target user $y$. Therefore, $r_{yj} = r_{xj} + d_{xy} = 1.5 - 0.25 = 1.25$.

**Algorithm 1** DSS algorithm for unknown rating prediction and Top-$k$ recommendation
```
1: Calculate similarity for all possible s_uv between user u and user v.
2: For each target user, the set of all his neighbors with similarity
   greater than a threshold (T_h = 0.0 in our experiments) is recorded
   (N_u).
3: Calculate distance between user u and each of his neighbor in N_u,
   d_uv standing for distance between user u and user v.
4: for each prediction target r_ui ∈ TestSet do
5:   Select all neighbors with distance d_uv for user u, who has rated
i.
6:   Predict r̃_ui by Eq 4.
7:   Add the predicted rating into predicted set, P ← P + r̃_ui
8: end for
9: Sort P by descending order of the predicted rating.
10: L ← Select top-k prediction in P as recommendation list.
11: return The predicted rating P and recommendation list L for u.
```

## Main equations of the proposed method

The similarity between users is calculated based on the unknown ratings in the training set. In general, the greater the similarity between the target user and his neighbor user, the higher the percentage of that neighbor in the prediction. We use Pearson's correlation coefficient to calculate the similarity between two users based on their common rating characteristics, which is defined as Eq 1.

$$s_{uv} = \frac{\sum_{i \in I_{uv}} (r_{ui} - \bar{r}_u)(r_{vi} - \bar{r}_v)}{\sqrt{\sum_{i \in I_{u \cap v}} (r_{ui} - \bar{r}_u)^2} \sqrt{\sum_{i \in I_{u \cap v}} (r_{vi} - \bar{r}_v)^2}},$$

(1)

where the set $I_{u \cap v}$ represents the collection of items that users $u$, $v$ shared, and $r_{ui}$ and $r_{vi}$ represent the ratings given by users $u$, $v$ on the same item $i$, respectively. $\bar{r}_u$ and $\bar{r}_v$ represent the average ratings of users $u$ and $v$, respectively.

Moreover, the rating distance between two users is defined by Eq 2:

$$d_{vu} = \sum_{i \in I_{uv}} \frac{r_{vi} - r_{ui}}{|I_{uv}|},$$

(2)

where distance $d_{vu}$ between users is equal to the mean of the differences of the one-to-one corresponding ratings in the set of items $I_{uv}$ jointly rated by the two users. $r_{ui}$ and $r_{vi}$ are the ratings of user $u$ and user $v$ on item $i$, and $|I_{uv}|$ is the number of the subset.

However, we only calculate user pairs with the similarity (Eq 1) larger or equal to the threshold, filtering them based on Eq 3, and consider the qualified users as the neighbors for

the target user.

$$
\begin{cases}
v \in N_u, & if \quad s_{(v,u)} \geq T_h \\
v \notin N_u, & if \quad s_{(v,u)} < T_h
\end{cases},
\tag{3}
$$

where $N_u$ is the selected neighbor set for user $u$, and $s_{(v, u)}$ is the similarity between user $u$ and $v$. $T_h$ is the threshold set by experiments.

On this basis, Eq 4 can be used to calculate the unknown rating of user $u$ on item $i$.

$$
\tilde{r}_{ui} = \frac{1}{|N_u|} \sum_{v \in N_u} (d_{vu} + r_{vi})
\tag{4}
$$

where $|N_u|$ is the neighbor number of user $u$, $d_{vu}$ is the rating distance between user $v$ and user $u$.

## Experiment

### Experimental data set

We use MoviesLens data set (ML-25M, download at https://grouplens.org/datasets/movielens/25m/) in our experiments, in which describes 5-star rating and free-text tagging activity from a movie recommendation service [52]. It contains 25000095 ratings and 1093360 tag applications across 62423 movies. The rating ranges from 0.5 to 5, with a 0.5 interval. The number of users in the data set is 162541, and we randomly selected 3000 of these users for the experiment to save computational resources and time. It contains 431783 ratings across 62423 movies, and we only use the rating data.

Additionally, 10-fold cross-validation approach is used to determine how generalizable the suggested strategy is. The data set is divided into ten similar-size groups at random. Each experiment uses one of the data sets as the test set, while the other nine sets are used as the training set. The average of all ten experiments was calculated as the final result.

### Benchmark algorithms

We compare the proposed DSS algorithm to average rating (AverageRating), Slope One algorithm, cosine similarity collaborative filtering (Cosine), resource allocation collaborative filtering (RA), user opinion spreading (UOS), and multi-level collaborative filtering (MLCF) to demonstrate the effectiveness of DSS. The main ideas of these algorithms used for comparison are described below.

1. The user's average rating (AverageRating) predicts the target user's rating based on his average rating on all other items. If there is no ratings of the traget user in the traning set, the method takes the average ratings of all other users as the result.

2. Cosine similarity collaborative filtering (Cosine) [53] is to find the similarity between two rating vectors by measuring the cosine value of the angle between them. The vectors contains ratings from two users on shared items.

3. Resource allocation collaborative filtering (RA) [54, 55] is to use the concept of link prediction in the recommendation system to improve performance. Popular products should have less impact when determining user similarity because most users like popular items, hence they are not suited for users to offer recommendations. RA is a popularity-based measure of local similarity, and its value for two users is determined by the degree of common scoring items between them. The lower the RA similarity index of a movie with a

common rating, the more popular it is. Furthermore, as the number of shared rated items grows, the value of RA grows, and the estimated similarity becomes more reliable.

4. User opinion spreading (UOS) [56] algorithm is a combination of collaborative filtering algorithm and user opinion dissemination process. If two users have the same positive or negative view on the same item, their opinions are consistent, and the great similarity between them indicates that their tastes are similar. On the other hand, the similarity is low if the two people have opposing viewpoints on the same items. In other words, users share their opinions on items that have been jointly assessed in the user-item bipartite graph model, and UOS measures user similarity based on the attitudes of the items that users have rated in common.

5. Multi-level collaborative filtering (MLCF) [57] algorithm is designed to improve the similarity method, divide the Pearson similarity into different levels, and impose different constraints, so as to improve the accuracy of the classic collaborative filtering algorithm.

## Performance criteria

We evaluate the design algorithm's performance in six dimensions. The similarity weights for filtering are established first. Second, we look at the error level the rating prediction. Third, after generating the user's Top-$k$ recommendation list, we examine the ranking performance of user's favorite items in the list. Forth, we evaluate the diversity of the recommendation lists. Fifth, the prediction of users' likes or dislikes is verified by comparison. Finally, we measure the scalability of the proposed algorithm by comparing the number of neighbors used in the prediction and the time spent on the computation.

**Similarity selection threshold.** According to Eq 3, each user has a group of neighbors at varying distances, and the degree of similarity between the user and his neighbors varies. Neighbors with a high degree of similarity are worthy of reference, while neighbors with a low degree of similarity not only provide no reference but also degrade prediction accuracy.

Therefore, we design experiments to decide the proper value of $T_h \in [-1.0, 1.0]$, as shown in Fig 2. The results suggest that the appropriate similarity threshold is $T_h = 0.0$, and the proposed DSS has the optimal prediction accuracy, with mean absolute error at 0.667 and root mean square error at 0.874. Although the accuracy improvement seems trivial in Fig 2, the number of neighbor used in the prediction is much lower (details presented in Fig 12).

**Measure errors in predicting the ratings.** Our work uses mean absolute error (*MAE*) [5], root mean square error (*RMSE*) [58] to evaluate the error of rating prediction. The *MAE* reflects the deviation between the algorithm's predicted ratings and the user's actual ratings, while the *RMSE* represents the root mean square value of the deviation between the predicted user ratings and the actual ratings.

It's important to note that the *MAE* (Eq 5) value represents the average errors, and the *RMSE* (Eq 6) adds up the square of errors before extraction of a root, increasing the error gap and penalizing the predicted value with large errors. The lower the result of these two parameters is, the smaller the prediction error and the better the algorithm prediction performance is.

$$MAE = \frac{\sum_{i=1}^{n} |r_{ui} - \tilde{r}_{ui}|}{n} \tag{5}$$

$$RMSE = \frac{\sqrt{\sum_{i=1}^{n} |r_{ui} - \tilde{r}_{ui}|^2}}{n} \tag{6}$$

where $n$ is the number of predictions in the testing set, the user's predicted score value is represented by $\tilde{r}_{ui}$, and the user's actual rating value is represented by $r_{ui}$.

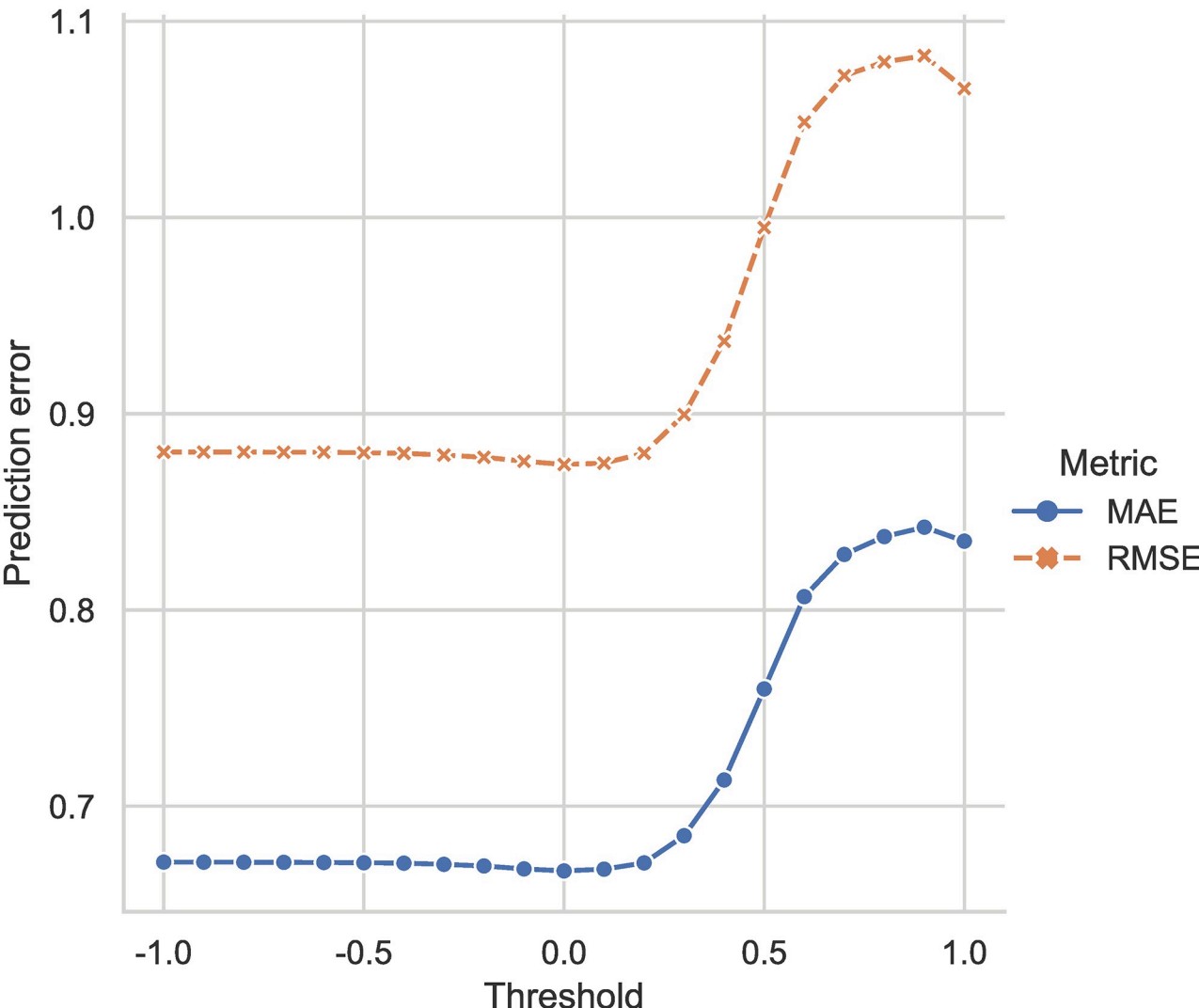

**Fig 2. Comparison of errors generated by different similarity selection thresholds under the same experimental conditions (mean absolute error and root mean square error on MovieLens with selected users).**

The *MAE* of all algorithms is shown in Fig 3. The *MAE* of the proposed DSS algorithm is 0.6647, which indicates the highest prediction accuracy and best performance compared to others. The Slope One has an *MAE* of 0.6707. The proposed DSS reduces errors by 0.9 percent. *MAE* of DSS has been improved by 2 percent on average when compared to all other algorithms.

As illustrated in Fig 4, the *RMSE* also shows the extent of the recommender system's prediction error. *RMSE*, in comparison to *MAE*, is more sensitive to larger errors. With a value of 0.8766, the DSS algorithm has the smallest result. The *RMSE* of the Slope One method is 0.8840, which is the second-best. The DSS lowers the error rate by 0.83 percent. *RMSE* of DSS has been improved by 2.3 percent on average as compared to all other algorithms.

**Ranking of Top-*k* recommendations.**   When making Top-*k* recommendations, it is not enough to predict the unknown user-item ratings; the recommendation system must also filter the user's favorite items based on the prediction results and generate a list of recommendations for these items in descending order of the user's preferences.

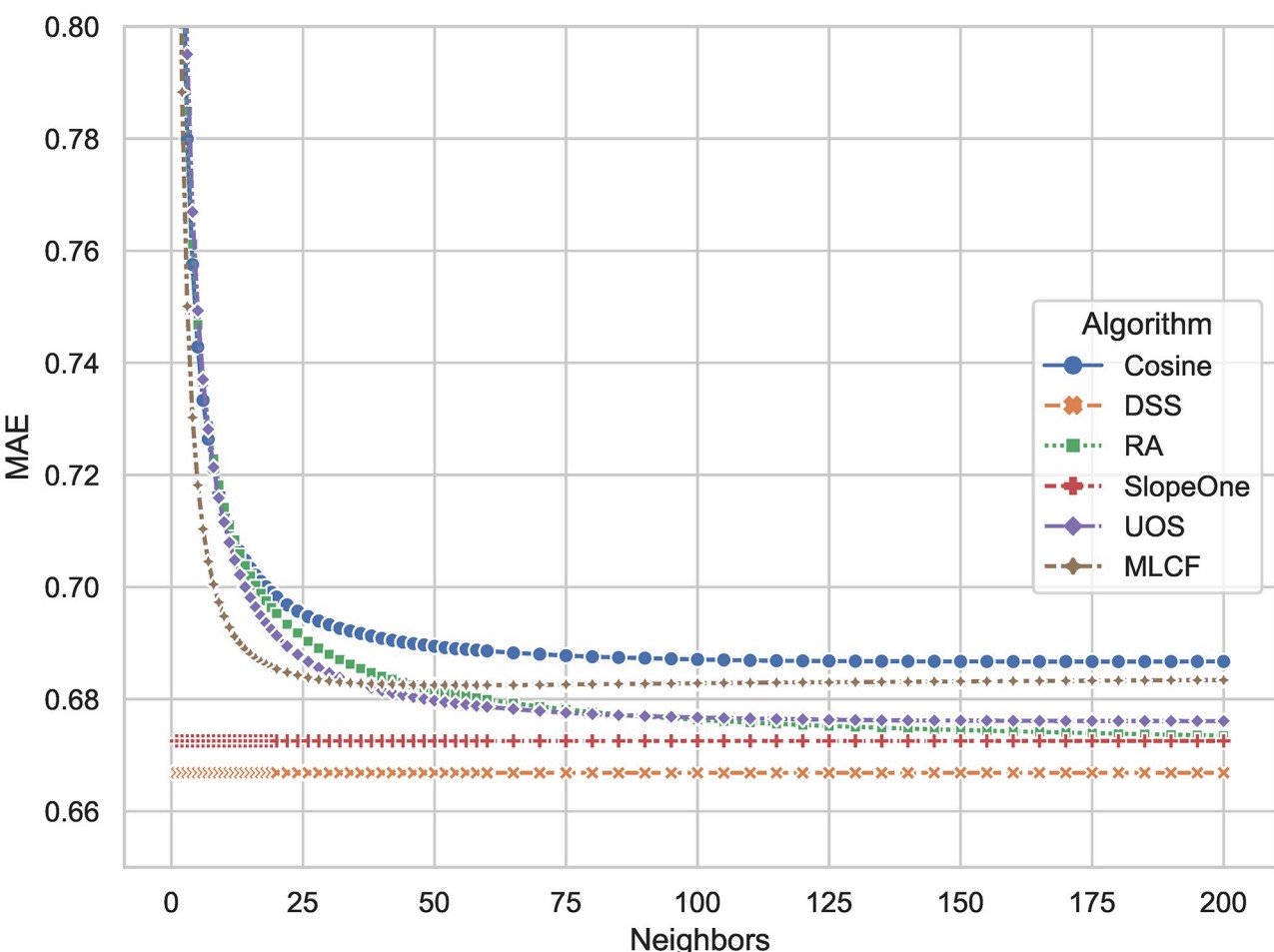

**Fig 3. *MAE* comparison results of DSS and other algorithms.** The smaller the result, the smaller the prediction error of the algorithms on the rating.

We utilize the average rating of each user in the training set as a threshold for whether a user likes or dislikes a certain item. For instance, suppose the average of a user $u$ in the training set is 3.5, then the user $u$ likes item $i$ when the rating $r_{ui}$ is 4.5 and user $u$ dislikes item $i$ if the rating $r_{ui}$ is 3. The recommender system selects those items that are predicted to be preferred by users and sorts them in descending order by predicted ratings.

The experimental approach in this paper dictates that the number of scored items is distributed differently for each user in each trial. If user $u$ has $k_u$ ratings on different items in the testing set for the first trial, the algorithm predicts each of the ratings, and only those with predicted ratings greater than the average rating of users in the training set are chosen as Top-$k$ suggested items.

In our work, we use half-life utility index (*HLU*) and normalized discounted cumulative gain (*NDCG*) to evaluate the ranking performance of Top-$k$ suggested items [5].

*HLU* (Eq 7) measures the practicability of the recommendation list for users.

$$HLU = \frac{\sum_{i=1}^{m} \frac{max(r_{ui} - d, 0)}{2^{(i-1)/(h-1)}}}{m} \tag{7}$$

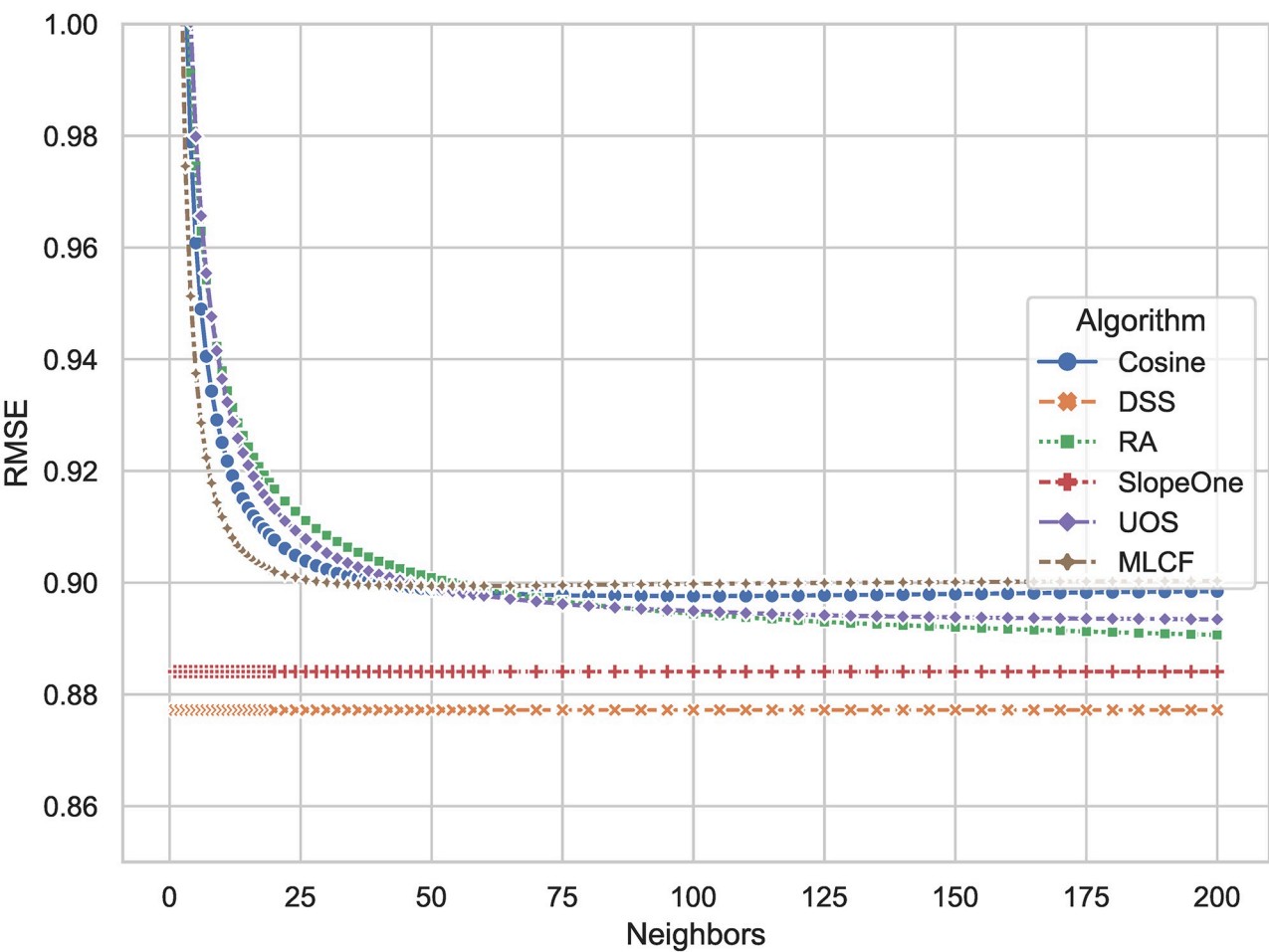

**Fig 4. *RMSE* comparison results of DSS and other algorithms.** The smaller the result, the smaller the prediction error of the algorithms on the rating.

where $m$ is the number of item in recommendation lists, $r_{ui}$ represents the rating of the user $u$ on the movie $i$, $d$ is the default rating (such as the average rating), and $h$ is the system's Half-life, the value of $h$ in our experiments is set as 2. The larger the value of *HLU*, the better the ranking of recommendation list.

*HLU* reflects the user's level of interest in the recommended list, as seen in Fig 5. The higher the score, the more interested the user is in the limited pages. The DSS method has a score of 1.050, while the Slope One algorithm had a score of 1.021, an increase of 2.8 percent. When compared to other algorithms, *HLU* has a 5.5 percent increase on average.

*NDCG* (Eq 9) reveals that the user's favorite movies being ranked first in the recommended list will increase the user's experience to a greater extent. The more relevant the items, the higher the ranking, the higher the user's satisfaction with the system.

$$DCG = \sum_{i=1}^{b} R_i + \sum_{i=b+1}^{N} \frac{R_i}{\log_b r_i} \qquad (8)$$

$$NDCG = \frac{\sum_{u=1}^{m} \frac{DCG}{max(DCG_u)}}{m} \qquad (9)$$

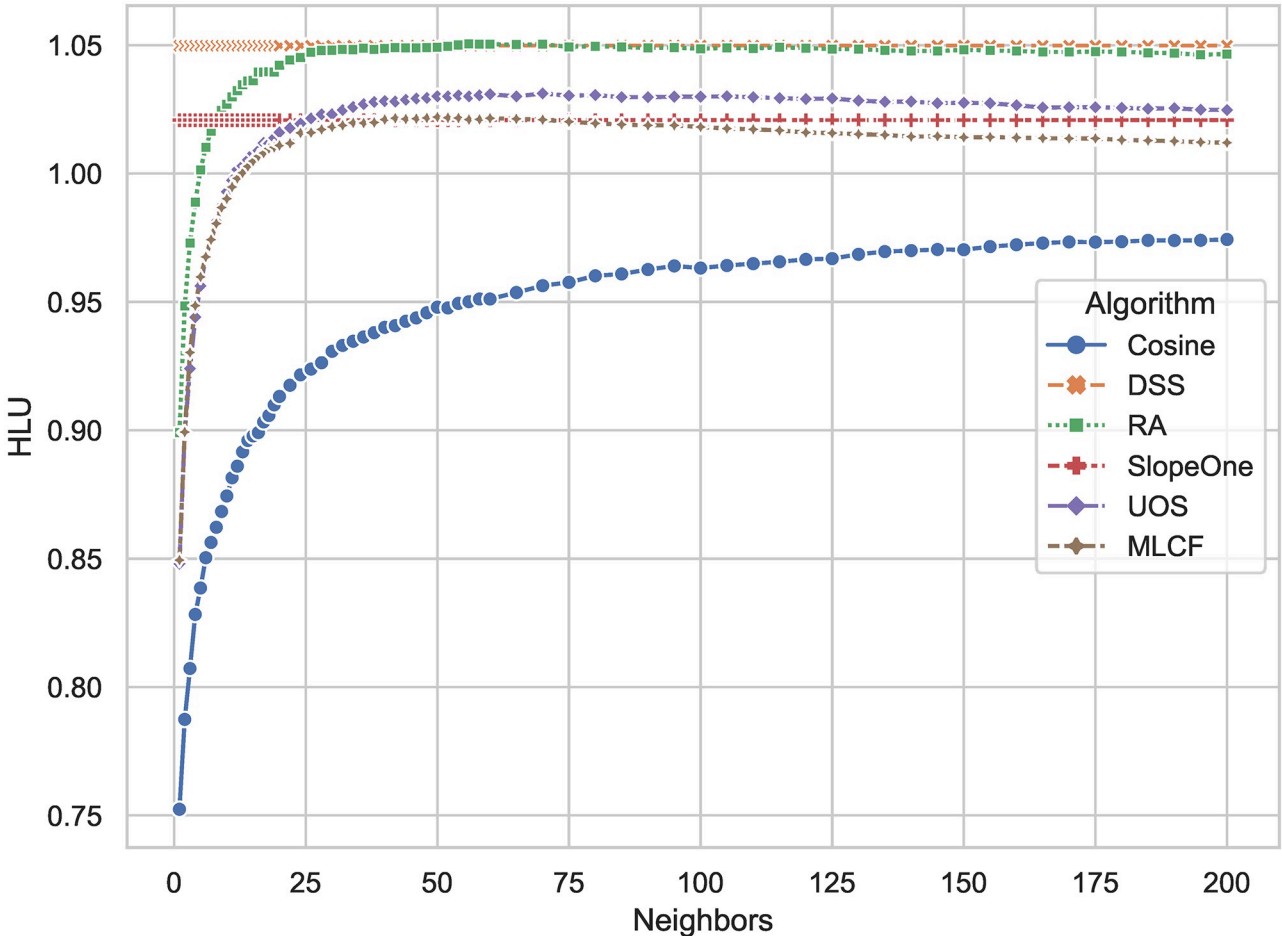

**Fig 5. HLU comparison results of DSS and other algorithms.** The larger the result, the higher the probability that the algorithm give a recommendation list with the items that the user likes at the top.

where *DCG* is calculated based on the list of real and predicted ratings of each user. $R_i$ indicates whether the movie ranked in *i* is liked by the user, $R_i = 1$ indicates that the user likes the movie, $R_i = 0$ means that the user does not like the movie. *NDCG* is the normalization of *DCG*, given by Eq 9, which normalizes the value range to between 0 and 1. A high *NDCG* usually indicates a favorable item suggestion order, as well as a higher ability of the algorithm to rank items in the list.

The results of *NDCG* is shown in Fig 6. Because the *NDCG* shows the recommendation list's ranking performance, and the higher the score, the better. The DSS method has the largest score of 0.755, which is about 1.3 percent higher than the Slope One algorithm, with a score of 0.745. When compared to other algorithms, *NDCG* has a 2.8 percent increase on average.

**Diversity of recommendations.** The average similarity between items in a recommendation list is commonly used in the domain to measure the diversity of a recommendation list, and the diversity of a suggestion list is beneficial to improve the user experience. According to this hypothesis, the smaller the average similarity of a recommendation list, the greater the diversity [5].

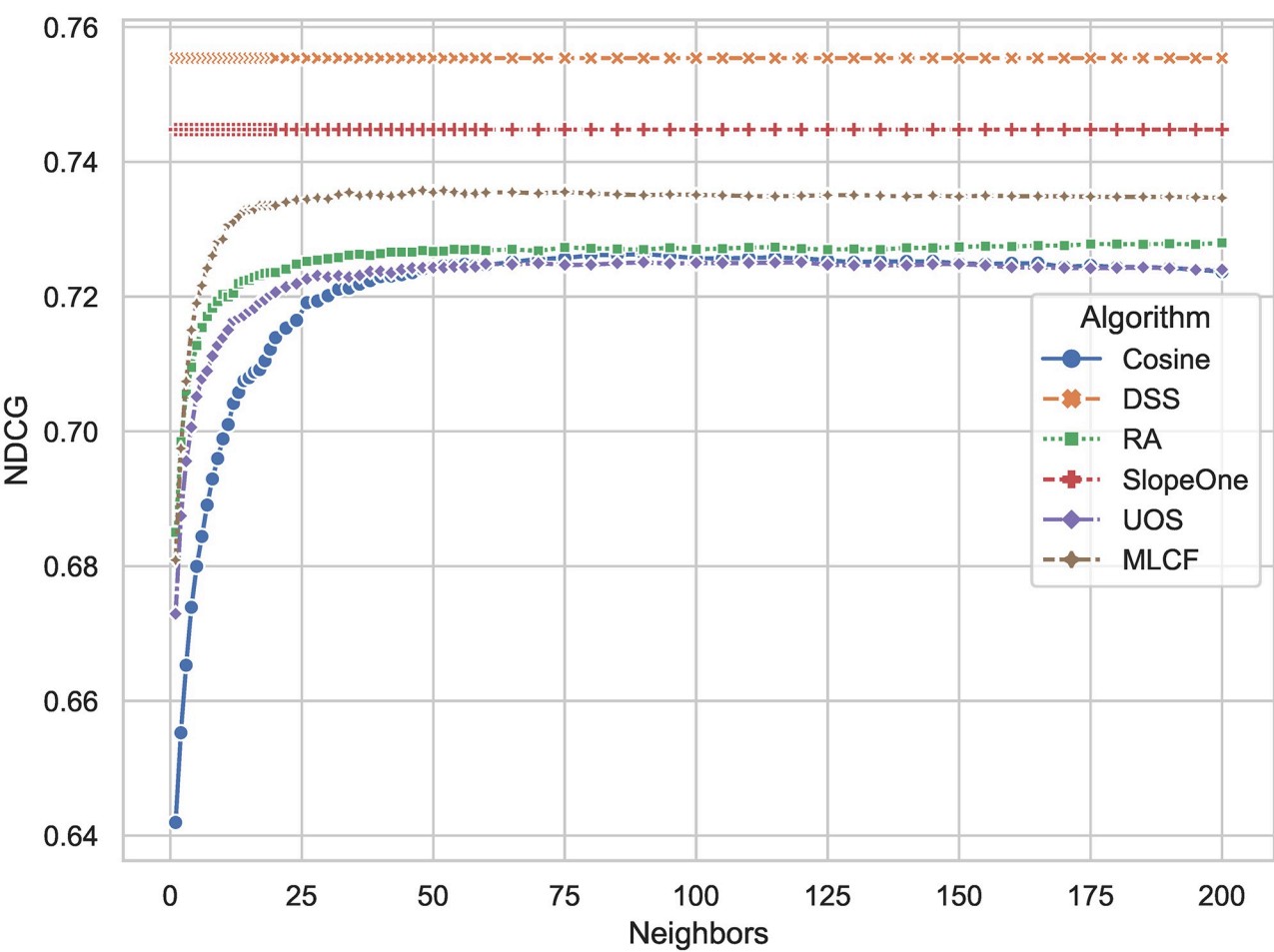

**Fig 6. *NDCG* comparison results of DSS and other algorithms.** The larger the result, the higher the probability that the algorithm give a recommendation list with the items that the user likes at the top.

In our work, we propose that a diverse recommendation list should include both low degree and high degree items because the degree of items in recommender systems is frequently used to measure their popularity [59].

Based on this assumption, the ratio of the standard deviation of the item degrees in a recommendation list to the mean of the item degrees can be used to measure the diversity of the recommendation list, i.e., which popular items and niche items are included in the recommendation list.

Therefore, the standard deviation dispersion ratio is proposed to measure the diversity of recommendation list. The average diversity of the recommendation list created for each user can be calculated using the Eqs 10 – 12 for a given training and test set.

$$Diversity = \frac{\sum_{l \in L} \left( \frac{\sigma(l)}{Ave(l)} \right)}{|L|}, \tag{10}$$

where $L$ is the set of recommendation lists for all users in the test set, $\sigma(l)$ is the standard

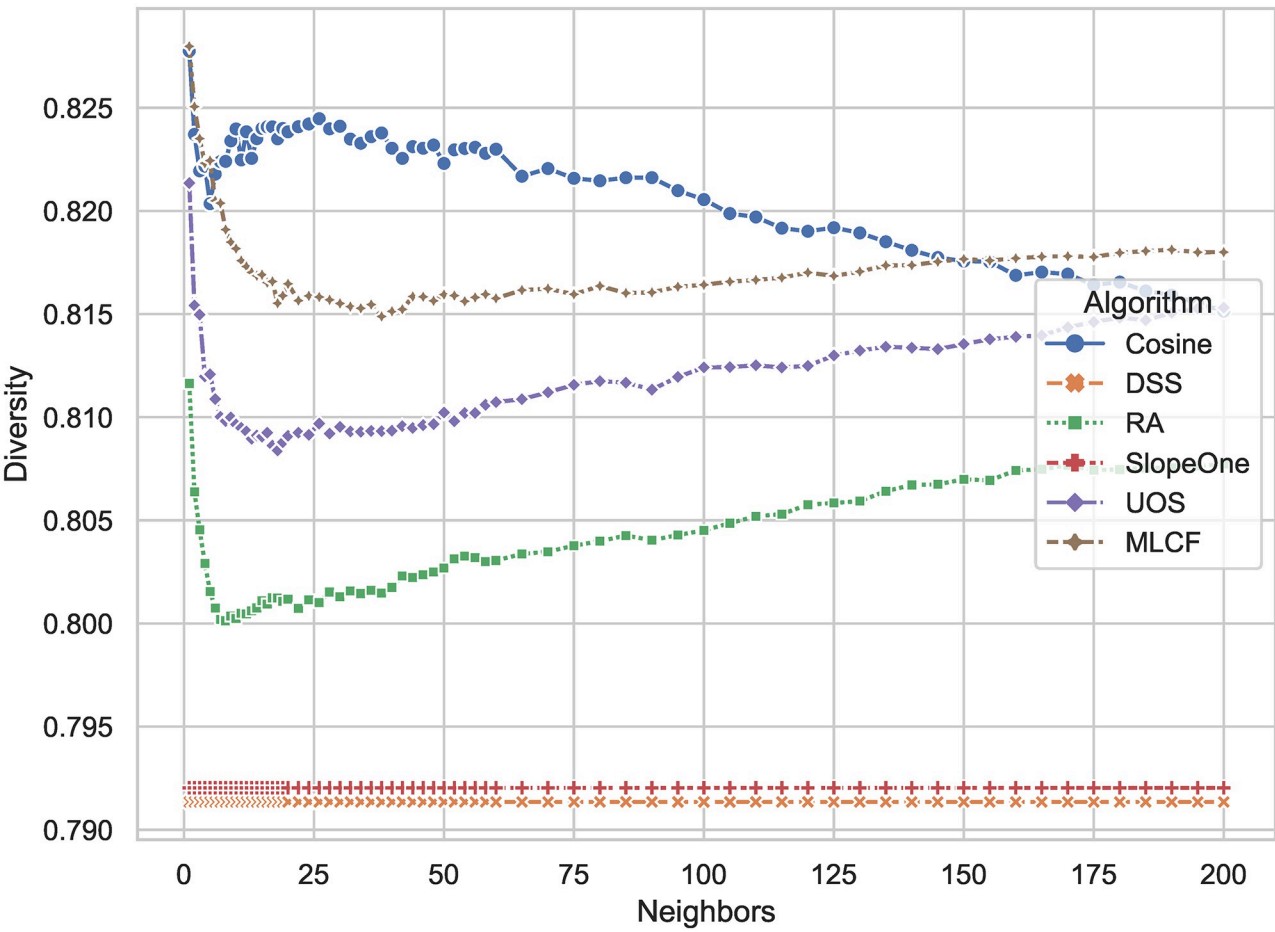

**Fig 7. Diversity comparison results of DSS and other algorithms.** The larger the result, the more disperse the degree distribution of items in the recommendation list given by the algorithm, and the better the diversity of the recommendation list.

deviation of item degree in a list $l$, and $|l|$ is the size of the list.

$$\sigma(l) = \sqrt{\frac{1}{|l|} \sum_{i=1}^{l} (k_i - \bar{k}_l)^2}, \tag{11}$$

$$Ave(l) = \frac{1}{|l|} \sum_{i=1}^{l} k_i, \tag{12}$$

where $i$ is an item in the recommendation list, $k_i$ is the degree of the item indicating the number of users has rated the item. $\bar{k}_l$ is the average degree of the items in the list.

The standard deviation dispersion ratio, as illustrated in Fig 7, indicates the diversity of the suggestion list. The more diversified the distribution of degrees in recommended lists, the higher the diversity rating. As can be observed from the graph, the proposed DSS has a value of 0.791, which is somewhat lower than all the other approaches. Slope One has a 0.1 percent higher as 0.792. The highest diversity is provided by Cosine CF, which is also the one with the largest prediction errors in Fig 3. The results highlight the frequent tension between prediction accuracy and recommendation diversity in recommender systems.

**Area under curve and accuracy.** When predicting if a user likes an item, as stated in the preceding section, the recommendation algorithm predicts that the user likes the item when

validated in the test set, and this situation is classified as a true positive (*TP*). Similarly, when a user likes an item and the recommendation algorithm predicts that the user does not like the item, this situation is known as a false negative (*FN*). When a user dislikes an item and the recommendation algorithm also predicts that the user does not like the item, this situation is known as a true negative (*TN*). When a user dislikes an item and the recommendation algorithm predicts that the user likes the item, this situation classified as a false positive (*FP*).

On this basis, Eqs 13 – 15, are given for measuring the prediction accuracy of user preference.

$$Precision = \frac{TP}{TP + FP} \tag{13}$$

$$Recall = \frac{TP}{TP + FN} \tag{14}$$

$$Accuracy = \frac{TP + TN}{TP + FP + FN + TN} \tag{15}$$

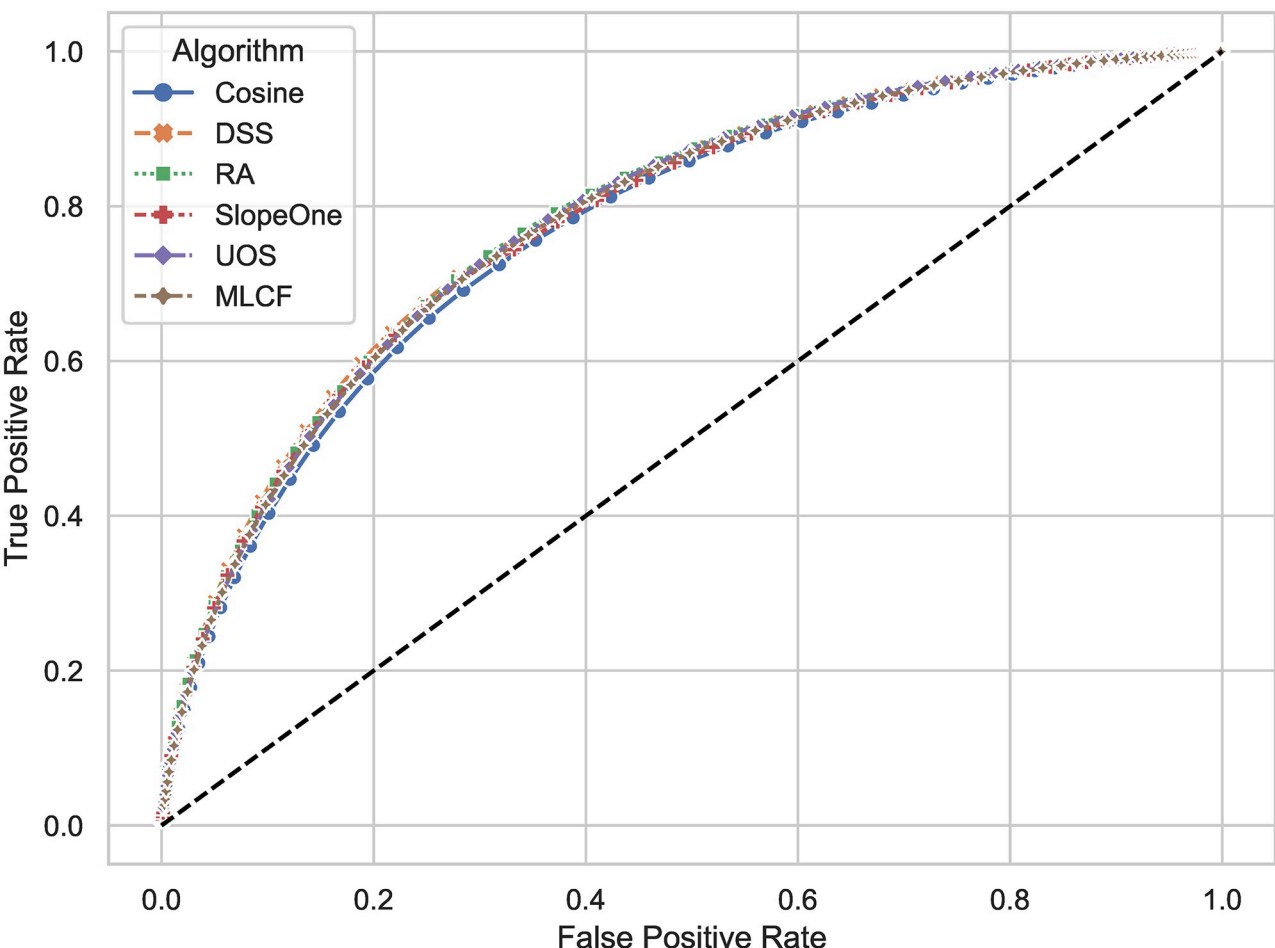

**Fig 8. Comparison of receiver operating characteristic curve (*ROC*).** The area under curve (*AUC*) of DSS is the largest between false positive rate 0.0 and 0.3, the third between 0.3 and 0.8375, and the second between 0.8375 and 1.0, but the advantage of leading algorithms at all stages is very small. For example, DSS has a 1.67 percent advantage over RA at FPR = 0.19 (false positive rate).

where *TP*, *FP*, *TN* and *FN* stand for the number of true positive, false positive, true negative and false negative, respectively.

By changing the threshold of the algorithm to determine whether a user likes an item or not, we can get the receiver operating characteristic curve (*ROC*) in Fig 8. The area under curve (*AUC*) of DSS is the largest between false positive rate 0.0 and 0.3, the third between 0.3 and 0.8375, and the second between 0.8375 and 1.0. The area of DSS is the second over all, second only to RA.

It is shown that Fig 9 shows that the *Precision* defined by Eq 13. DSS has *Precision* = 0.7107, compared with Slope One with *Precision* = 0.7060, improving 0.67 percent. By analyzing the definition, it can be concluded that the DSS algorithm has less false positive cases in the testing set than other comparative algorithms, and DSS is also with low prediction error and higher ranking accuracy.

The comparison of *Recall* is presented in Fig 10, which is defined by Eq 14. DSS has *Recall* = 0.7470, compared with Slope One with *Recall* = 0.7454, deteriorating 0.21 percent. By contrast to *Precision*, it can be concluded that the DSS algorithm has more false negative cases in the testing set than RA, UOS and MLCF. The recall of DSS indicates the algorithm tends to predict ratings lower than the actual ones, as RA and UOS tend to predict ratings higher.

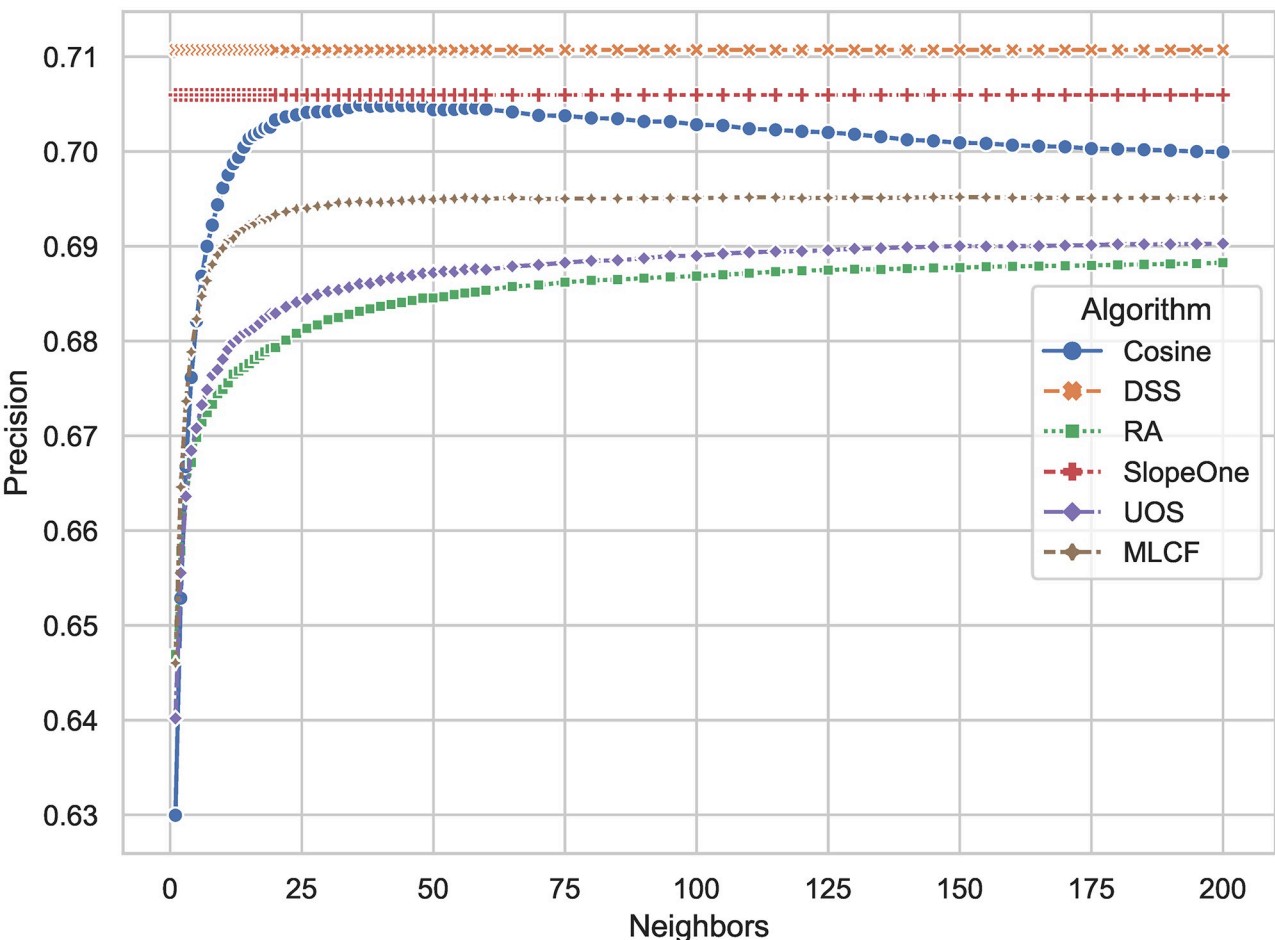

**Fig 9. *Precision* results of different algorithms.** The larger the result, the less false positives the algorithm has in predicting user preferences, and the less likely the algorithm is to put items that the user does not like into the recommendation list.

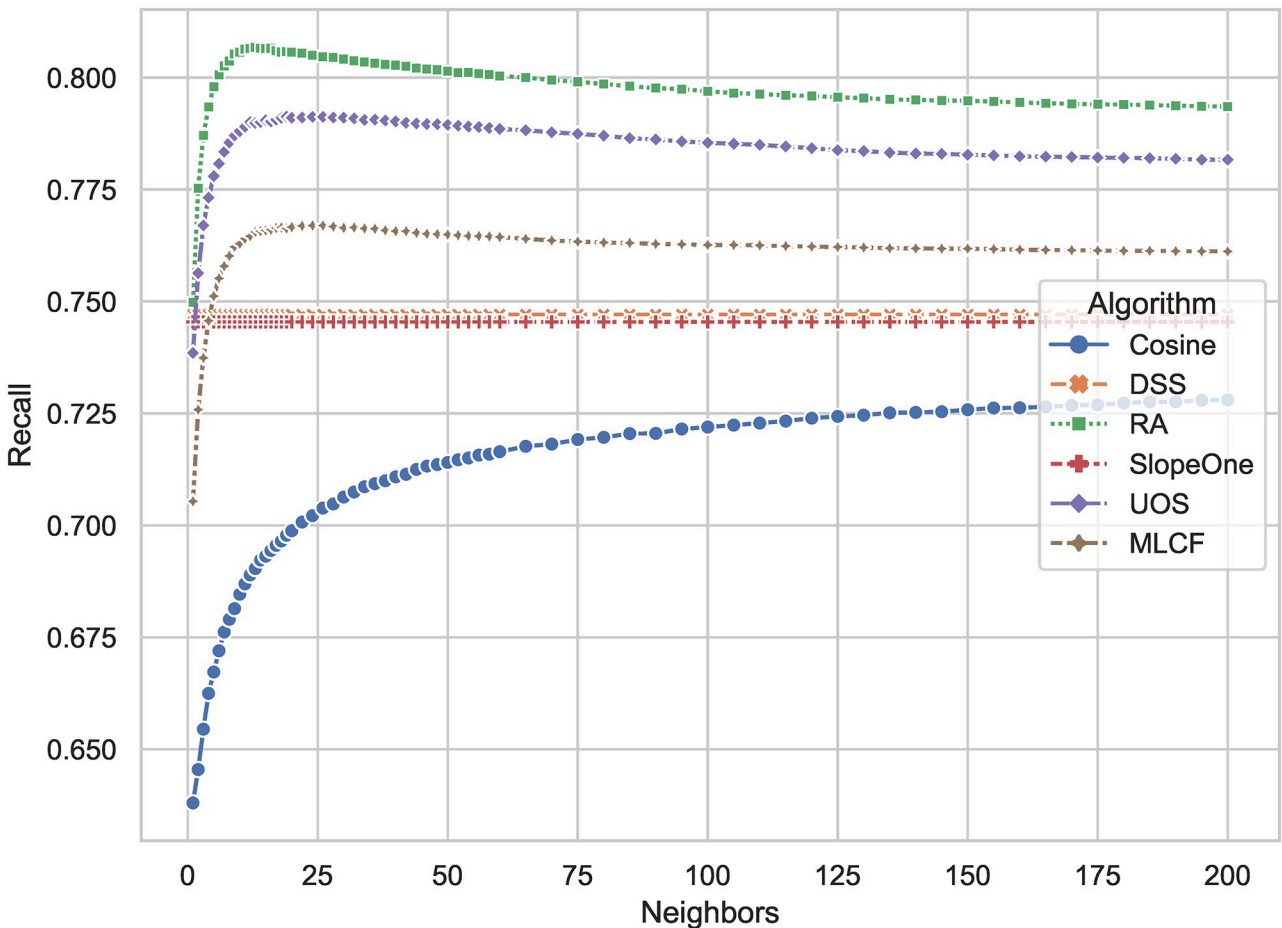

**Fig 10. *Recall* results of different algorithms.** The larger the result, the less false negatives the algorithm has in predicting the user's preferences, and the less likely the algorithm will give a recommendation list that leaves out the items the user likes.

The results of the *Accuracy* is presented in Fig 11, which is defined by Eq 15. DSS has *Accuracy* = 0.7165, compared with Slope One with *Accuracy* = 0.7124, improving 0.68 percent. By analyzing the definitions, it can be concluded that the DSS algorithm has a higher overall combined accuracy in predicting user preferences.

**Scalability of algorithms.** Scalability generally considers the ability of different algorithms to adapt in the face of increasing amounts of data in real applications. We explore the number of neighbor used in prediction and the time used in the calculation of the method. Because the number of neighbors relied on at prediction is often cached up in the actual system, the fewer neighbors required the more storage space is saved in real applications.

Figs 12 and 13 show the number of neighbors used in the prediction against the required number of neighbors in algorithm parameters. We test both 3000 randomly selected users and 1000 randomly selected users. Compared to classical distance-based algorithm, the number of neighbors used by DSS is reduce by 29.6 percent and 26.5 percent, respectively. When there are 1000 users in the data set, DSS can do the prediction with less neighbors than all other algorithms. The lower the number of users in the training set, the lower the number of neighbors used by the DSS. In practical applications, the DSS algorithm can further improve the scalability of the algorithm by limiting the number of users involved in the calculation.

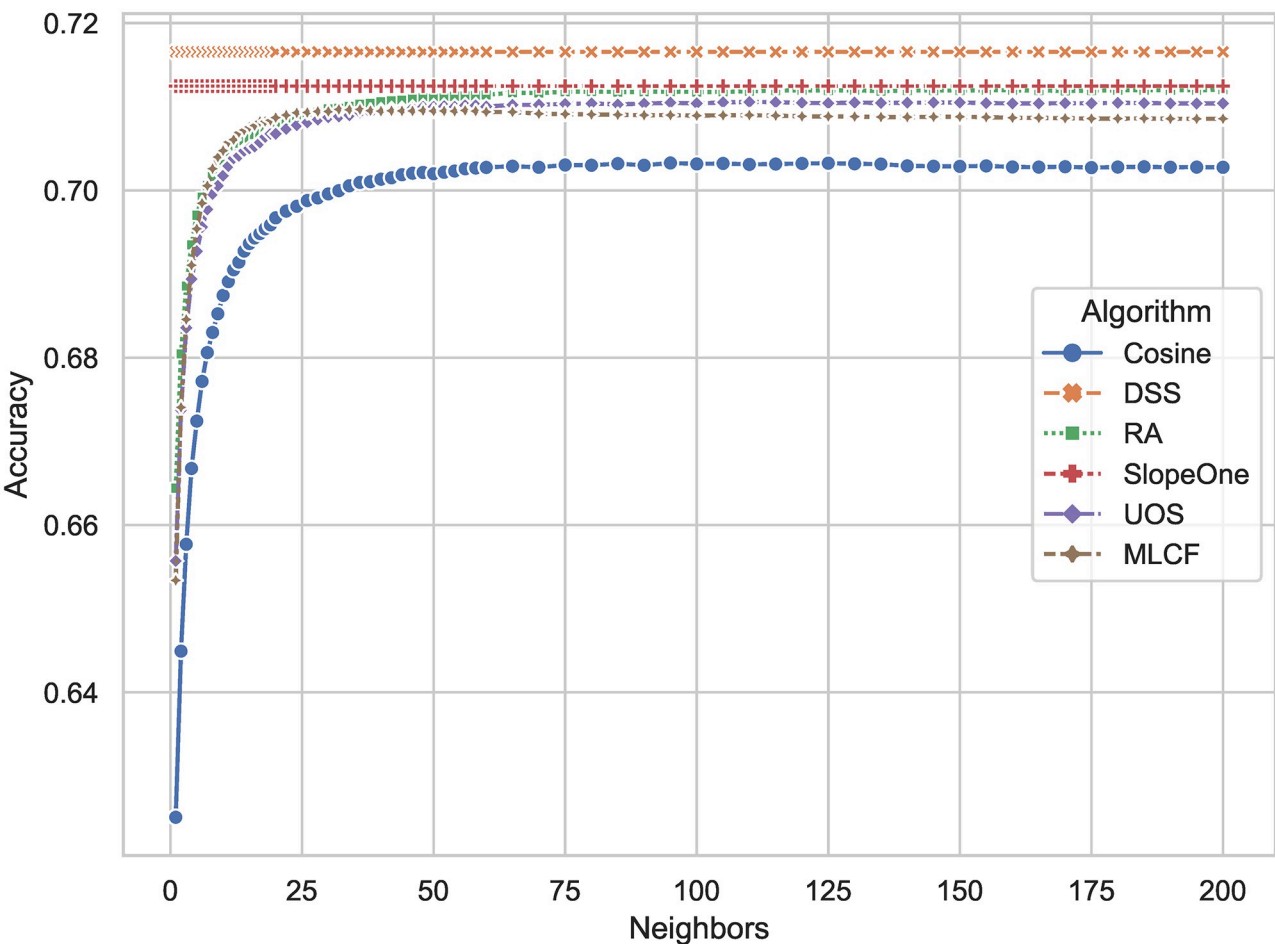

**Fig 11. *Accuracy* of different algorithms.** The larger the result, the more accurate the algorithm is in predicting user preferences.

The DSS algorithm, which is the second fastest of the six approaches, can be finished in a very short period of time due to the simplicity of distance-based methods and the similarity selection before prediction. The DSS calculation employs substantially fewer neighbors after employing similarity to filter out those less important neighbors, resulting in a DSS calculation time that is even faster than Slope One. The detailed results is shown in Table 1.

## Conclusions

To overcome the problem that distance-based recommendation algorithms like Slope One use too many neighbors in prediction, we combine distance-based and similarity-based collaborative filtering algorithms. In this study, users with similarity below a threshold are removed from the set of neighbors of the predicted target, and the study examines in detail the selection of thresholds for user rating distance and user similarity in recommender systems. Using the inter-user rating distance to estimate the target users' ratings of unknown items, a novel algorithm for user-item link prediction in recommender systems is devised based on the similarity selection.

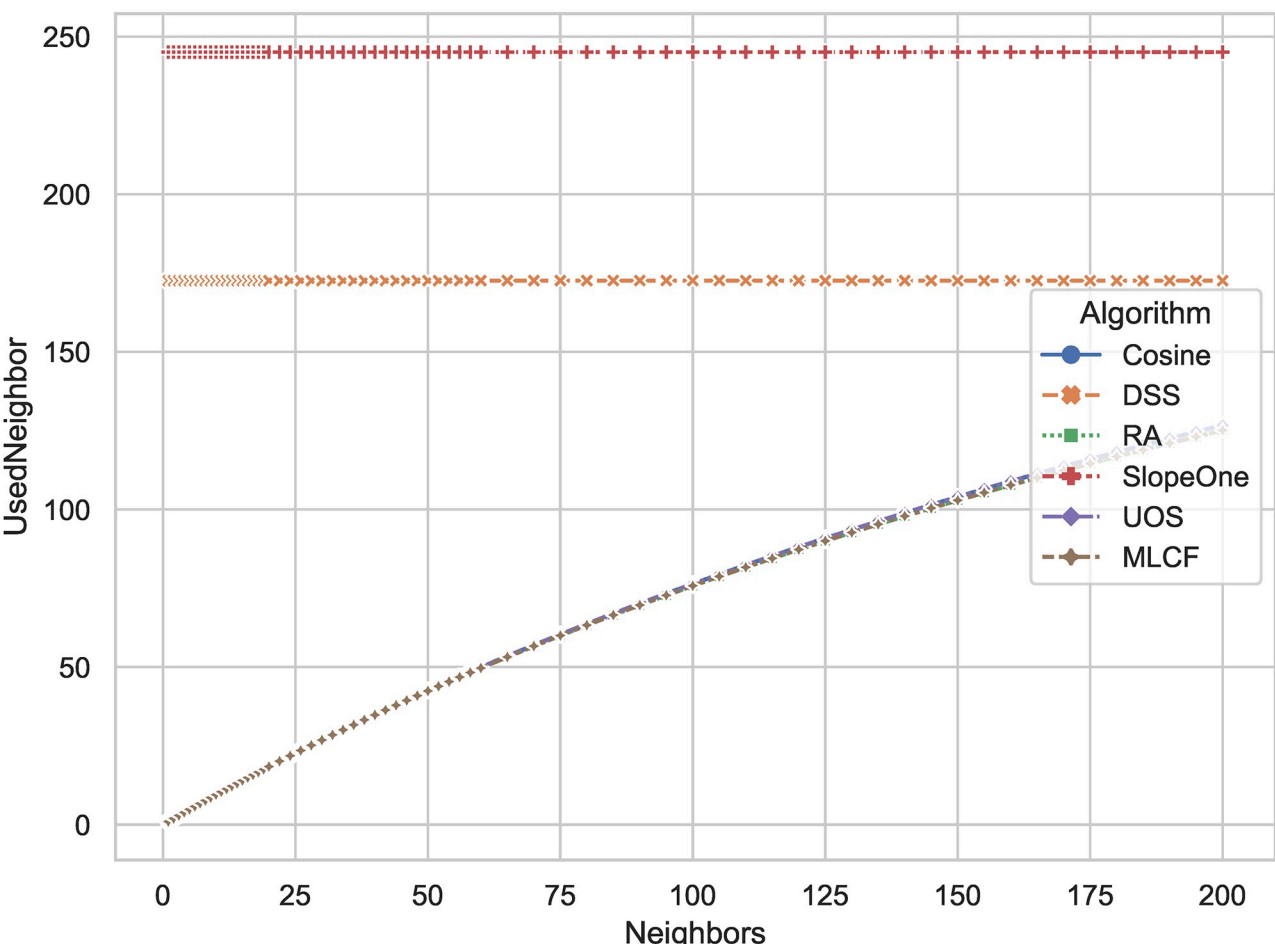

**Fig 12. The actual number of neighbors used in prediction (y-axis) against the number of neighbors that algorithms plan to select (x-axis).** The number of users selected for experiments is 3000.

In comparison to the original algorithm, the proposed method reduces the computational complexity, the number of neighbors required in the prediction process, the prediction errors, and improves the rationality of the recommendation list ranking by 1 percent. Based on our understanding, the main reason why the algorithm designed in this paper can improve prediction accuracy and recommendation performance is that negatively correlated neighbors can only play a negative role in the prediction, so removing these neighbors has a positive effect on both prediction and recommendation.

However, it is evident in the experiments that DSS, the distance model-based algorithm designed in this paper, improves the prediction accuracy and reduces the number of neighbors used leading to a decrease in its recommendation diversity and recall rate.

Therefore, our work can continue to be explored in the future in the following ways. The first possible option is to consider about how to improve recall for better preference prediction. The second choice is to consider other information of the data set can be used in distance-based model. The last option is to study the effect of system features such as the long-tail distribution of ratings on prediction.

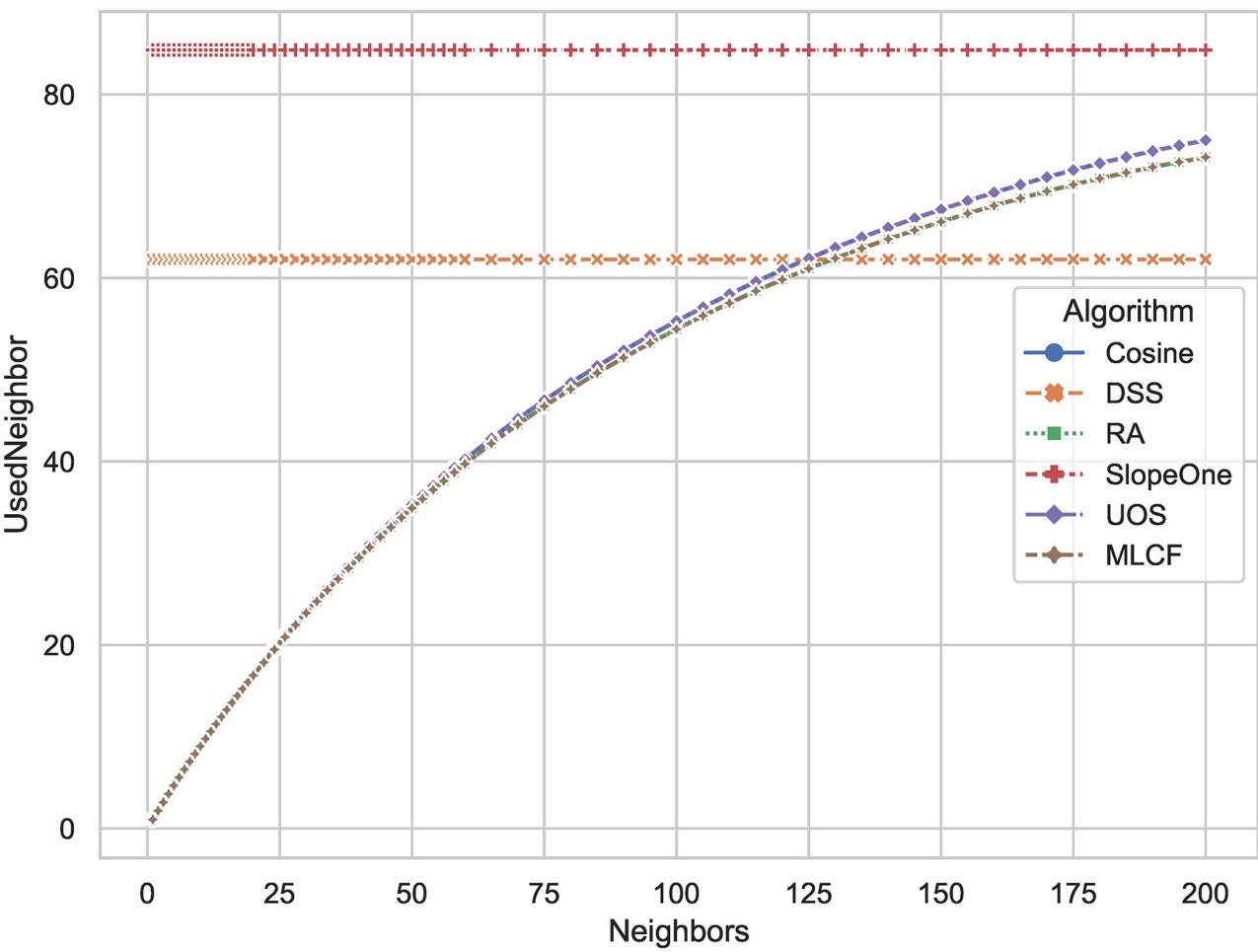

**Fig 13. The actual number of neighbors used in prediction (y-axis) against the number of neighbors that algorithms plan to select (x-axis).** The number of users selected for the experiment is 1000.

**Table 1. The time it takes to compute each method (in minutes) with identical hardware and parameters.**

| Algorithm | DSS | Consine | UOS | RA | MLCF | SlopeOne |
|---|---|---|---|---|---|---|
| Time(min.) | 4.388 | 8.180 | 8.525 | 8.663 | 7.750 | 4.057 |

## Supporting information

**S1 File.**
(ZIP)

## Acknowledgments

Zhan Su and Jun Ai would like to express their love to Lingyi Ai and thank her for inspiring us to keep fighting.

## Author Contributions

**Conceptualization:** Jun Ai.

**Formal analysis:** Zhong Huang.

**Funding acquisition:** Zhan Su.

**Methodology:** Zhan Su.

**Project administration:** Zhan Su.

**Resources:** Xuanxiong Zhang, Fengyu Zhao.

**Software:** Zhong Huang, Jun Ai.

**Validation:** Zhong Huang, Jun Ai.

**Writing – original draft:** Zhong Huang, Jun Ai.

**Writing – review & editing:** Zhan Su, Jun Ai, Xuanxiong Zhang, Lihui Shang, Fengyu Zhao.

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
