## [Decision Letter · Decision Letter 0]

4 Nov 2021

PONE-D-21-27341Link prediction in recommender systems based on user-rating distance and similarity selectionPLOS ONE

Dear Dr. Ai,

Thank you for submitting your manuscript to PLOS ONE. After careful consideration, we feel that it has merit but does not fully meet PLOS ONE’s publication criteria as it currently stands. Therefore, we invite you to submit a revised version of the manuscript that addresses the points raised during the review process.

We look forward to receiving your revised manuscript.

Kind regards,

Hua Wang

Academic Editor

PLOS ONE

Journal Requirements:

"The research project is supported by the Young Scientists Fund of the National Natural Science Foundation of China (Grant No. 61803264). Zhan Su and Jun Ai would want to convey their love and appreciation to Lingyi Ai, Zhan Su’s daughter, for inspiring us to keep fighting"

"Zhan Su received the funding by the Young Scientists Fund of the National Natural Science Foundation of China (Grant No. 61803264).

The funder' website is at http://www.nsfc.gov.cn/.

The funders had no role in study design, data collection and analysis, decision to publish, or preparation of the manuscript"

3. We note that you have stated that you will provide repository information for your data at acceptance. Should your manuscript be accepted for publication, we will hold it until you provide the relevant accession numbers or DOIs necessary to access your data. If you wish to make changes to your Data Availability statement, please describe these changes in your cover letter and we will update your Data Availability statement to reflect the information you provide

Additional Editor Comments (if provided):

One paragraph of the motivation of the paper with research questions is needed. That will attract readers in future.

Reviewers' comments:

Reviewer's Responses to Questions

**Comments to the Author**

1. Is the manuscript technically sound, and do the data support the conclusions?

Reviewer #1: Yes

Reviewer #2: Partly

2. Has the statistical analysis been performed appropriately and rigorously? 

Reviewer #1: Yes

Reviewer #2: Yes

3. Have the authors made all data underlying the findings in their manuscript fully available?

Reviewer #1: Yes

Reviewer #2: Yes

4. Is the manuscript presented in an intelligible fashion and written in standard English?

Reviewer #1: Yes

Reviewer #2: No

5. Review Comments to the Author

Reviewer #1: The authors present a user-item link prediction approach that combines user distance measurement with similarity-based user selection, and the experimental results show the model achieves satisfied performance. However, some questions should be question or solved:

1.Page 2, line 46, what does it means by “reducing the sparsity of the Slope One method.”

2.The authors mentioned some technical problems, but feel a little mixed, if authors can highlight two or three technical difficulties, and in the performance evaluation targeted to show the advantages of the proposed method, it would be better.

3.The main contribution of this article is mentioned on page 3, but the contribution should be linked to the requirements, as well as the problems that these contributions can answer.

4.Have the authors tried to change the similarity-selection equation (3) to other forms? For example, use |s(u,v)|>=T_h instead of s(u,v). Negative correlation is likely to be helpful in prediction.

5.Authors need to add more discussion on why the similarity selection improves the overall performance.

Reviewer #2: The paper provides a link prediction method based on user distance and similarity selection. Some issues should be further considered in this work;

- Refer to Algorithm 1, please provide more discussion on top-k prediction in recommendation list, in your work. Please give more explanation on it in the experimental results.

- Could you please add AUC of the results in the experiments.

-Please discuss in more detail the information you have employed in your work. As you know, there are some information in MovieLens. Have you only used rating rate?

-Does the method help to solve the sparsity problem? Furthermore, there is a long tailed distribution for number of ratings. How does the proposed method deal with ling tailed ratings?

It is more interesting to see the pitfalls of the method and future line of this study in Conclusion section.

6. PLOS authors have the option to publish the peer review history of their article (what does this mean?). If published, this will include your full peer review and any attached files.

Reviewer #1: No

Reviewer #2: No

---

## [Author Response · Author response to Decision Letter 0]

5 Feb 2022

Reviewer#1, Concern # 1: Page 2, line 46, what does it means by “reducing the sparsity of the Slope One method.”

Author response: Thank you for the comment. The main problem addressed in this paper is the scalability (the number of used neighbors in prediction) and prediction accuracy of model-based collaborative filtering. We have further elaborated on this goal by modifying the abstract and contribution sections.

Author action: We have deleted the inaccurate expressions in the revised paper.

Reviewer#1, Concern # 2: The authors mentioned some technical problems, but feel a little mixed, if authors can highlight two or three technical difficulties, and in the performance evaluation targeted to show the advantages of the proposed method, it would be better.

Author response: Thank you for the comment. The main technical problem addressed in this paper is the scalability (the number of used neighbors in prediction) and prediction accuracy of model-based collaborative filtering. We have further elaborated on this goal by modifying the abstract and contribution sections.

Author action: In accordance with the reviewers' comments, we have revised the relevant expressions and marked them in the revised manuscript. On Page 3, marked yellow.

Reviewer#1, Concern # 3: The main contribution of this article is mentioned on page 3, but the contribution should be linked to the requirements, as well as the problems that these contributions can answer. 

Author response: 

Thank you for the comment. 

Author action: As the reviewer suggested, we have revised the contribution and marked them in the revised manuscript. On Page 3, marked yellow.

Reviewer#1, Concern # 4: Have the authors tried to change the similarity-selection equation (3) to other forms? For example, use |s(u,v)|>=T_h instead of s(u,v). Negative correlation is likely to be helpful in prediction.

Author response: 

Thank you for the comment. We considered several similar techniques of filtering during the experimental exploratory phase, but the results were not promising. The findings of a typical comparative experiment are shown in the following figures (see attached file includes MAE and NDCG, the method reviewer suggested labeled as DSS-abs, the one we use in the paper is labeled as DSS), which illustrates that filtering neighbors with large absolute values retains both positively and negatively correlated neighbors (|s(u,v)|>=T_h), but will cause the prediction error to become larger and NDCG ranking is little worse than the original Slope One. Thus, we did not include other forms of experimental results in the revised manuscript. 

Author action: None

Reviewer#1, Concern # 5: Authors need to add more discussion on why the similarity selection improves the overall performance.

Author response: 

Thank you for the comment. 

Author action: 

As suggested by the reviewer, we've added a corresponding discussion and analysis in the conclusion section. On Page 15 and 16, marked yellow.

Reviewer#2, Concern # 1: Refer to Algorithm 1, please provide more discussion on top-k prediction in recommendation list, in your work. Please give more explanation on it in the experimental results.

Author response: 

Thank you for the comment. 

Author action: 

As suggested by the reviewer, we have added discussion of the Top-k prediction in the revised paper. On Page 9, marked yellow.

Reviewer#2, Concern # 2: Could you please add AUC of the results in the experiments.

Author response: 

Thank you for the advice. 

Author action: 

As suggested by the reviewer, we've added Recall, Precision, Accuracy and ROC/AUC in the revised paper. On Page 12,13 and 14, add Fig 8, 9, 10 and 11, marked yellow.

Reviewer#2, Concern # 3: Please discuss in more detail the information you have employed in your work. As you know, there are some information in MovieLens. Have you only used rating rate?

Author response: 

Thank you for the comment. We use only the ratings data from the MovieLens data set.

Author action: 

We have added discussion in the revised paper to address all the detail of experiments. We also increased the number of users used in the experiments, and redrew completely new figures of the experiments. On Page 7, marked yellow.

Reviewer#2, Concern # 4: Does the method help to solve the sparsity problem? Furthermore, there is a long tailed distribution for number of ratings. How does the proposed method deal with ling tailed ratings?

Author response: 

Thank you for the advice. 

Our main concern addressed in this paper is the scalability of model-based collaborative filtering, the proposed method aims to reduce the number of required neighbors of prediction, as well as improving the prediction accuracy. Thus, the method doesn’t help with the sparsity problem.

Additionally, we don't currently consider the long-tail distribution of scores, but we'd would like to look into it in the future. 

Author action: 

We've included the suggestion in the conclusion section, along with a forecast for future study, and we're grateful to the reviewers for his or her insightful advice. On Page 16, marked yellow.

Reviewer#2, Concern # 5: It is more interesting to see the pitfalls of the method and future line of this study in Conclusion section.

Author response: 

Thank you for the advice. The drawback of the algorithm is that the computational steps are added, but the reduction of the prediction error is not significant, and the similarity in the distance-based prediction process is not utilized in the previous calculation.

Author action: Based on the reviewers' suggestions, we have added a relevant discussion in the conclusion section. On Page 16, marked yellow.

---

## [Decision Letter · Decision Letter 1]

31 May 2022

PONE-D-21-27341R1Enhancing the scalability of distance-based link prediction algorithms in recommender systems through similarity selectionPLOS ONE

Dear Dr. Ai,

Thank you for submitting your manuscript to PLOS ONE. After careful consideration, we feel that it has merit but does not fully meet PLOS ONE’s publication criteria as it currently stands. Therefore, we invite you to submit a revised version of the manuscript that addresses the points raised during the review process.

We look forward to receiving your revised manuscript.

Kind regards,

Hua Wang

Academic Editor

PLOS ONE

Journal Requirements:

Reviewers' comments:

Reviewer's Responses to Questions

**Comments to the Author**

1. If the authors have adequately addressed your comments raised in a previous round of review and you feel that this manuscript is now acceptable for publication, you may indicate that here to bypass the “Comments to the Author” section, enter your conflict of interest statement in the “Confidential to Editor” section, and submit your "Accept" recommendation.

Reviewer #1: All comments have been addressed

Reviewer #2: All comments have been addressed

2. Is the manuscript technically sound, and do the data support the conclusions?

Reviewer #1: Yes

Reviewer #2: Yes

3. Has the statistical analysis been performed appropriately and rigorously? 

Reviewer #1: Yes

Reviewer #2: Yes

4. Have the authors made all data underlying the findings in their manuscript fully available?

Reviewer #1: Yes

Reviewer #2: Yes

5. Is the manuscript presented in an intelligible fashion and written in standard English?

Reviewer #1: Yes

Reviewer #2: Yes

6. Review Comments to the Author

Reviewer #1: The authors have revised this manuscript carefully and stated how they revised in detail as the reviewers suggested, and I am satisfied with this revision in principle. The manuscript also includes a few writing mistakes, and I suggest the authors carefully to proofreading this manuscript.

+ Line 67 on Page 10: The statement of "Not only is the Slope One method simple to implement, but it is also highly effective” is suggested to be revised to "The Slope One method is not only simple to implement, but also highly effective".

+The citations of figures in this manuscript are inconsistent, such as "Fig 2", "Figure 3", "figure 3", etc.

+The metrics, such as NDCG (Line 290 on Page 18), DCG (Line 295 on Page 18), should be italic.

In summary, this paper could be accepted after minor revise.

Reviewer #2: (No Response)

7. PLOS authors have the option to publish the peer review history of their article (what does this mean?). If published, this will include your full peer review and any attached files.

Reviewer #1: No

Reviewer #2: No

---

## [Author Response · Author response to Decision Letter 1]

5 Jun 2022

Reviewer#1, Comment# 1: The authors have revised this manuscript carefully and stated how they revised in detail as the reviewers suggested, and I am satisfied with this revision in principle. The manuscript also includes a few writing mistakes, and I suggest the authors carefully to proofreading this manuscript.

Author response: Thank you for the comment. We have carefully proofread the article. 

Author action: Writing errors and typos have been corrected in the revised paper, which were marked yellow in the marked manuscript.

Reviewer#1, Comment # 2: Line 67 on Page 10: The statement of "Not only is the Slope One method simple to implement, but it is also highly effective” is suggested to be revised to "The Slope One method is not only simple to implement, but also highly effective".

Author response: Thank you for the comment.

Author action: The issues pointed out by the reviewers have been corrected in the revised manuscript.

Reviewer#1, Concern # 3: The citations of figures in this manuscript are inconsistent, such as "Fig 2", "Figure 3", "figure 3", etc.

Author response: Thank you for the comment. 

Author action: The issues pointed out by the reviewers have been corrected in the revised manuscript.

Reviewer#1, Concern # 4: The metrics, such as NDCG (Line 290 on Page 18), DCG (Line 295 on Page 18), should be italic.

Author response: Thank you for the comment.

 Author action: The issues pointed out by the reviewers have been corrected in the revised manuscript.

Reviewer#2,

No further comments

---

## [Decision Letter · Decision Letter 2]

11 Jul 2022

Enhancing the scalability of distance-based link prediction algorithms in recommender systems through similarity selection

PONE-D-21-27341R2

Dear Dr. Ai,

We’re pleased to inform you that your manuscript has been judged scientifically suitable for publication and will be formally accepted for publication once it meets all outstanding technical requirements.

Kind regards,

Hua Wang

Academic Editor

PLOS ONE

Additional Editor Comments (optional):

The new version has been improved the paper quality. Great efforts!

Reviewers' comments:

Reviewer's Responses to Questions

**Comments to the Author**

1. If the authors have adequately addressed your comments raised in a previous round of review and you feel that this manuscript is now acceptable for publication, you may indicate that here to bypass the “Comments to the Author” section, enter your conflict of interest statement in the “Confidential to Editor” section, and submit your "Accept" recommendation.

Reviewer #1: All comments have been addressed

2. Is the manuscript technically sound, and do the data support the conclusions?

Reviewer #1: Yes

3. Has the statistical analysis been performed appropriately and rigorously? 

Reviewer #1: Yes

4. Have the authors made all data underlying the findings in their manuscript fully available?

Reviewer #1: Yes

5. Is the manuscript presented in an intelligible fashion and written in standard English?

Reviewer #1: Yes

6. Review Comments to the Author

Reviewer #1: The authors have revised all the mistakes and answered all the questions, and I have no further comments.

7. PLOS authors have the option to publish the peer review history of their article (what does this mean?). If published, this will include your full peer review and any attached files.

Reviewer #1: No

---

## [Editor Report · Acceptance letter]

19 Jul 2022

PONE-D-21-27341R2 

Enhancing the scalability of distance-based link prediction algorithms in recommender systems through similarity selection 

Dear Dr. Ai:

I'm pleased to inform you that your manuscript has been deemed suitable for publication in PLOS ONE. Congratulations! Your manuscript is now with our production department. 

Kind regards, 

on behalf of

Dr. Hua Wang 

Academic Editor

PLOS ONE